# Intuitionistic Fuzzy Entropy for Group Decision Making of Water Engineering Project Delivery System Selection

**DOI:** 10.3390/e21111101

**Published:** 2019-11-11

**Authors:** Xun Liu, Fei Qian, Lingna Lin, Kun Zhang, Lianbo Zhu

**Affiliations:** 1School of Civil Engineering, Suzhou University of Science and Technology, Suzhou 215000, China; linlingna1010@126.com (L.L.); lbzhu@usts.edu.cn (L.Z.); 2Institute of Engineering Management, Hohai University, Nanjing 211100, China; qf2019@hhu.edu.cn (F.Q.); dreamerzk@126.com (K.Z.)

**Keywords:** water engineering, project delivery system, intuitionistic fuzzy entropy, group decision making

## Abstract

The project delivery mode is an extremely important link in the life cycle of water engineering. Many cases show that increases in the costs, construction period, and claims in the course of the implementation of water engineering are related to the decision of the project delivery mode in the early stages. Therefore, it is particularly important to choose a delivery mode that matches the water engineering. On the basis of identifying the key factors that affect the decision on the project delivery system and establishing a set of index systems, a comprehensive decision of engineering transaction is essentially considered to be a fuzzy multi-attribute group decision. In this study, intuitionistic fuzzy entropy was used to determine the weight of the influencing factors on the engineering transaction mode; then, intuitionistic fuzzy entropy was used to determine the weight of decision experts. Thus, a comprehensive scheme-ranking model based on an intuitionistic fuzzy hybrid average (IFHA) operator and intuitionistic fuzzy weighted average (IFWA) operator was established. Finally, a practical case analysis of a hydropower station further demonstrated the feasibility, objectivity, and scientific nature of the decision model.

## 1. Introduction

For a water engineering project, the construction process is essentially the process of exchanges between the owner and contractor to obtain a construction product, but the particularity of the project itself makes the project delivery different from general commodity transactions, as engineering projects must rely on a certain mode of delivery. The project delivery system (PDS) not only defines the roles and responsibilities of each participant in the project, but also determines the payment method of the owner and the risk allocation of each participant, which provides a framework for the organization and implementation of the project [1]. Several PDSs that can be selected for the owner including design–bid–build (DBB), design–build (DB), construction management at risk (CM-at risk), or construction management as general contractor (GC), engineering–procurement–construction (EPC), and integrated project delivery (IPD) [2,3,4,5,6]. Each model of PDS has its own characteristics and requirements. There is not a universal PDS suitable for all types of water engineering projects under different situations. The decision making of PDS is an important link in the whole life cycle of a water engineering project, and an appropriate PDS can effectively improve the project performance [7,8]. Therefore, choosing and tailoring the most appropriate needs of the PDS to customers is a crucial task in early stage of any water engineering project [9].

Previous researchers have carried out extensive studies on the choice and decision of PDS. Molenaar et al. [10] established five regression models through multiple regression to predict the cost, duration, expected consistency, management responsibility, and overall user satisfaction of the DB model, respectively. However, the model has strict hypothetical conditions and poor prediction accuracy, which was difficult to apply and popularize in practice. Khalil introduced his decision of PDS into an analytic hierarchy process (AHP) to establish the AHP decision model of PDS [11]; Mafakheri et al. combined the AHP method with a rough set, compared PDS by the AHP method, and sorted the scheme by the rough set method [12]. However, the AHP method relies too much on the subjective judgment of experts and has high uncertainty. In view of the difficulty of predicting PDS performance, historical case experience plays an important role in PDS decision making. Therefore, case-based reasoning (CBR) was also applied to the selection decision of PDS [13,14,15]. Luu et al. [13,15] established an index system of PDS decisions, and constructed the PDS decision support system based on case experiences. In this system, the owner can retrieve the case by calculating the similarity of the water engineering project. Nevertheless, the uniqueness of water engineering projects makes it difficult for two completely similar water engineering projects to exist, and the standard has to be simplified in CBR analysis, which greatly reduces the effectiveness of decision making. Lo et al. [16] used data envelopment analysis (DEA) to analyze the performance index efficiency of DBB, DB, CM, and design build maintain (DBM) in highway projects. This method has the advantage of objectivity, but it cannot directly solve the decision-making problem of PDS. Ling et al. [17] analyzed the index correlation based on data from an actual case, and used the artificial neural network (ANN) method to construct the project performance prediction model under the DB model, but it is difficult to obtain the index data of this method in practical application. Chen et al. [18] proposed a PDS selection decision model combining DEA and ANN based on the advantages of a DEA-BND (Bound Variable) model. Oyetunji et al. [8] put forward the fuzzy simple multi-attribute rating technique with swing weights (SMARTS) selection method for project delivery.

Overall, these methods are helpful to solve the problem of the PDS decision to a certain extent, but the choice of project delivery mode is often made intuitively according to the past experience and knowledge of decision makers, as well as the information and data of the water engineering project. The inherent law between the decision-making attribute and the transaction mode was not explored. In addition, most of the studies on the decision-making model were based on the subjective scoring and evaluation of experts due to the different experience of experts, different knowledge background, and the uniqueness and one-off characteristics of water engineering projects. Therefore, there is great subjectivity in the determination of attribute weight and expert weight, which would eventually lead to the deviation of decision results, affecting the implementation of water engineering projects. 

Group decision-making systems such as peer-to-peer (P2P) systems can be easily modeled as Fuzzy logic [19,20,21]. This opened a new area of research in decision making utilizing fuzzy sets (FSs) starting with Type-1 FSs, Type-2 FSs, and finally with intuitionistic fuzzy sets [20]. The comprehensive decision for project delivery is essentially a fuzzy multi-attribute group decision [21]; many researchers have done a lot work on fuzzy decision making for PDS selection [5,21,22,23,24], but the characteristics and fuzziness of the expert group were not considered yet. Therefore, it is necessary to select the proper PDS selection method to avoid the existing deficiencies. Studies have shown that fuzzy entropy is one of the most effective decision weights methods [25,26]. In this paper, based on fuzzy entropy theory, a group decision-making model to support PDS selection is proposed. In order to reduce the information lost in the overall judgment and improve the objectivity and fairness of group decision-making, the intuitionistic fuzzy hybrid average (IFHA) operator and intuitionistic fuzzy weighted average (IFWA) operator on the PDS decision are addressed to calculate the final decision weights. The present study aims to develop a more accurate and reliable PDS selection method. The main parts of this paper are organized as follows: Section 2 lists 15 key factors that affect the decision of water engineering PDS through a literature review, and establishes a set of index systems; Section 3 dealt with preliminaries; Section 4 presents the fuzzy group decision model for the selection of PDS; Section 5 presents a practical water engineering project case study that illustrates the applicability of this method; Section 6 provides conclusions. 

## 2. Influencing Factors Indexes of PDS

An analysis of influencing factors on a water engineering PDS is the basis of scientific decision-making, which is also a hot spot in the theoretical research on water engineering project delivery mode. Scholars have carried out extensive studies on the influencing factors of PDS through theoretical analysis and case analysis. However, water engineering projects have the characteristics of a large investment scale, long construction cycle, and many uncertain factors, so there should be many influencing factors in the choice of PDS. At present, a unified index system of influencing factors for PDS selection has not been formed yet; however, it is obvious that although there are differences in the emphasis and quantity of the existing PDS index system, the main influencing factors of PDS selection can be summarized into three categories of owner characteristics, project characteristics, and external environment, with a total 15 indicators, which include owner liability, owner participation, the owner’s own ability, risk allocation, owner design control, project scale, project complexity, project type, project scope clarity, project flexibility, project disputes, market competition, accessibility of materials, availability of technology, and the impact of laws and regulations. The details are shown in Table 1.

## 3. Preliminaries

In this section, the concept of intuitionistic fuzzy theory and a series of relevant algorithms are introduced.

Fuzzy set theory was first proposed by Zadeh (1965) [60] and has been widely used. The idea of a fuzzy set is to extend the eigenfunction with a value of only 1 or 0 to a membership function with arbitrary values in the unit closed interval [0, 1]. However, this kind of membership function value is only a single number; thus, the meaning of approval, disapproval, or hesitation cannot be expressed. Due to the fuzziness and uncertainty of objective things, Atanasov (1986) [61] extended the Zadeh fuzzy set to consider the membership, the non-membership degree, and the hesitation degree at the same time. Entropy, originally a thermodynamic unit, was later applied to information theory in 1940, and is a measure for uncertainty. Greater entropy represents more uncertain information in the single evaluation result of the decision expert; thus, a smaller weight should be given to such an expert. The cross entropy based on fuzzy set theory was proposed by Shang and Jiang [62] to describe the difference between two fuzzy sets. Vlachos and Sergiadis [26] put forward the cross-entropy of the intuitionistic fuzzy set (IFS) to describe the degree of difference between the intuitionistic fuzzy sets. When *x* and *y* are two discrete distributions, the relative entropy can be a measure of the degree of coincidence [63]. Some basic concepts and definitions of IFSs are presented as follows:

**Definition** **1**[61,64]**.**
*Let x be a non-empty set. An IFS_A_ is an object having the form:*
(1)$$A=\left\{\langle x,\mu_{A}(x),v_{A}(x)\rangle |x\in X\right\}$$
*where the mapping is presented as “*$\mu_{A}:X\to [0,1]$*” and “*$v_{A}:X\to [0,1]$*” under the condition “*$0\leq \mu_{A}(x)+v_{A}(x)\leq 1$*” for each*
x∈X*.*
$\mu_{A}(x)$
*and*
$v_{A}(x)$
*are defined as the degree of membership and the degree of non-membership, respectively, of element*
x∈X
*to set*
A.
*Obviously, if*
$v_{A}(x)=1-\mu_{A}(x)$
*, every*
IFS(A)
*on a non-empty set*
X
*becomes a fuzzy set.*

*In a more simple way, Xu and Yager*
*[65]*
*regarded*
$\alpha =(\mu_{\alpha },v_{\alpha })$
*as an intuitionistic fuzzy number and used it to represent an intuitionistic fuzzy set, where,*
$\mu_{\alpha }\in [0,1]$
*,*
$v_{\alpha }\in [0,1]$
*,*
$\mu_{\alpha }+v_{\alpha }\leq 1$
*.*


**Definition** **2**[61,64,66]**.**
*Let*
$\alpha =(\mu_{\alpha },v_{\alpha })$
*and*
$\beta =(\mu_{\beta },v_{\beta })$
*as any two intuitionistic fuzzy numbers; then, the algorithm for intuitionistic fuzzy numbers is as follows:*
*(1)* $\bar{\alpha }=(v_{\alpha },\mu_{\alpha })$*(2)* $\alpha ⊕\beta =(\mu_{\alpha }+\mu_{\beta }-\mu_{\alpha }\mu_{\beta },v_{\alpha }v_{\beta })$*(3)* $\lambda \alpha =(1-{\left(1-\mu_{\alpha }\right)}^{\lambda },v_{\alpha }^{\lambda }),\lambda >0$*(4)* $\alpha ⊗\beta =(\mu_{\alpha }\mu_{\beta },v_{\alpha }+v_{\beta }-v_{\alpha }v_{\beta })$*(5)* $\alpha^{\lambda }=(\mu_{\alpha }^{\lambda },1-{\left(1-v_{\alpha }\right)}^{\lambda }),\lambda >0$


**Definition** **3**[66]**.**
*As for the two intuitionistic fuzzy numbers:*
$\alpha_{1}=(\mu_{\alpha_{1}},v_{\alpha_{1}})$
*and*
$\alpha_{2}=(\mu_{\alpha_{2}},v_{\alpha_{2}})$*,*
$s(\alpha_{1})=\mu_{\alpha_{1}}-v_{\alpha_{1}}$
*and*
$s(\alpha_{2})=\mu_{\alpha_{2}}-v_{\alpha_{2}}$
*are the scoring functions of*
$\alpha_{1}$
*and*
$\alpha_{2}$
*respectively;*
$h(\alpha_{1})=\mu_{\alpha_{1}}+v_{\alpha_{1}}$
*and*
$h(\alpha_{2})=\mu_{\alpha_{2}}+v_{\alpha_{2}}$
*are the exact function of*
$\alpha_{1}$
*and*
$\alpha_{2}$
*respectively, so:*
*(1)* *if*$s(\alpha_{1})<s(\alpha_{2})$*, then*$\alpha_{1}<\alpha_{2}$*;**(2)* *if*$s(\alpha_{1})=s(\alpha_{2})$*, there are three situations:**(a)* $h(\alpha_{1})=h(\alpha_{2})$*, then*$\alpha_{1}=\alpha_{2}$*;**(b)* $h(\alpha_{1})<h(\alpha_{2})$*, then*$\alpha_{1}<\alpha_{2}$; *(c)* $h(\alpha_{1})>h(\alpha_{2})$*, then*$\alpha_{1}>\alpha_{2}$*.*

**Definition** **4**[67]**.**
*Let*
$x_{i},y_{i}\geq 0,i=1,2,\ldots ,n$*, and*
$1=\sum_{i=1}^{n}x_{i}\geq \sum_{i=1}^{n}y_{i}$*, then:*(2)$$h\left(X,Y\right)=\sum_{i}^{n}x_{i}\mathrm{lg}(x_{i}/y_{i})$$
*The above equation was described as the relative entropy of X relative to Y*
*, where,*
$X=(x_{1},x_{2},\ldots ,x_{n})$
*,*
$Y=(y_{1},y_{2},\ldots ,y_{n})$
*. Hence, relative entropy can be used to measure the degree of compliance between X and Y.*


**Definition** **5**[66]**.**
*The intuitionistic fuzzy mixed mean (IFHA) operator is a mapping**:*
$\Theta^{n}\to \Theta$*, it makes*
$IFHA_{\omega ,w}(\alpha_{1},\alpha_{2},\cdots ,\alpha_{n})=w_{1}{\dot{\alpha }}_{\sigma (1)}⊕w_{2}{\dot{\alpha }}_{\sigma (2)}⊕\cdots ⊕w_{n}{\dot{\alpha }}_{\sigma (n)}$*, where*
$w={\left(w_{1},w_{2},\ldots ,w_{n}\right)}^{T}$
*is the IFHA operator weight vector,*
$w_{j}\in [0,1](j=1,2,\ldots ,n)$*,*
$\sum_{j=1}^{n}w_{j}=1$*.*
${\dot{\alpha }}_{j}=n\omega_{j}\alpha_{j}(j=1,2,\ldots ,n)$*,*
$\left({\dot{\alpha }}_{\sigma (1)},{\dot{\alpha }}_{\sigma (2)},\ldots ,{\dot{\alpha }}_{\sigma (n)}\right)$
*weighted intuitionistic fuzzy array*
$\left({\dot{\alpha }}_{1},{\dot{\alpha }}_{2},\ldots ,{\dot{\alpha }}_{n}\right)$
*a replacement, it makes*
${\dot{\alpha }}_{\sigma (j)}\geq {\dot{\alpha }}_{\sigma (j+1)}(j=1,2,\ldots ,n-1)$*,*
$\omega ={\left(\omega_{1},\omega_{2},\ldots ,\omega_{n}\right)}^{T}$
*as*
$\alpha_{j}(j=1,2,\ldots ,n)$
*weight vector,*
$\omega_{j}\in [0,1](j=1,2,\ldots ,n)$*,*
$\sum_{j=1}^{n}\omega_{j}=1$*,*
n
*is the balance factor. Let*
${\dot{\alpha }}_{\sigma (j)}=(\mu_{{\dot{\alpha }}_{\sigma (j)}},v_{{\dot{\alpha }}_{\sigma (j)}}),(j=1,2,\cdots ,n)$*, then:*(3)$$IFHA_{\omega ,w}(\alpha_{1},\alpha_{2},\ldots ,\alpha_{n})=\left(1-\prod_{j=1}^{n}{\left(1-\mu_{{\dot{\alpha }}_{\sigma (j)}}\right)}^{w_{j}},\prod_{j=1}^{n}{\left(v_{{\dot{\alpha }}_{\sigma (j)}}\right)}^{w_{j}}\right)$$

**Definition** **6**[66]**.**
*Let*
$\alpha_{j}=\left(\mu_{\alpha_{j}},v_{\alpha_{j}}\right),\left(j=1,2,\cdots ,n\right)$
*be a set of intuitionistic fuzzy numbers, and let*
$IFWA:\Theta^{n}\to \Theta$*, if*
$IFWA_{\omega }(\alpha_{1},\alpha_{2},\ldots \alpha_{n})=\omega_{1}\alpha_{1}⊕\omega_{2}\alpha_{2}⊕\ldots ⊕\omega_{n}\alpha_{n}$*; then, IFWA is named the intuitionistic fuzzy weighted average operator, among them,*
$\omega ={\left(\omega_{1},\omega_{2},\ldots \omega_{n}\right)}^{T}$*is an exponential weight vector of*
$\alpha_{j}\left(j=1,2,\cdots ,n\right),\omega_{j}\in \left[0,1\right],\sum_{j-=1}^{n}\omega_{j}=1$*.*

**Definition** **7**[68]**.**
*The defined*
H:A→[0,1]
*was the entropy of IFS**:*
$A=\{<x,\mu_{A}(x),v_{A}(x)>|x\in X\}$*, therefore, the following can be calculated:*(4)$$H(A)=-\frac{1}{n\ln2}\sum_{i=1}^{n}\left[\mu_{A}(x)\ln\mu_{A}(x)+v_{A}(x)\ln v_{A}(x)-(1-\pi_{A}(A))\times \ln(1-\pi_{A}(A))-\pi_{A}(A)\ln2\right]$$

## 4. Establishment of Intuitionistic Fuzzy Group Decision Model

### 4.1. Group Decision Model Description

Set up *S* experts in the group: $D=\{D_{1},D_{2},\ldots ,D_{s}\}$ is an expert set, $M=\{M_{1},M_{2},\ldots ,M_{n}\}$ and $C=\{C_{1},C_{2},\ldots ,C_{m}\}$ are a set of schemes and decision attributes, respectively. The weight vector given to the experts by the subjective weighting method is $\xi ={\left(\xi_{1},\xi_{2},\ldots ,\xi_{s}\right)}^{T}$, $0\leq \xi_{k}\leq 1,k=1,2,\ldots ,s$ and $\sum_{k=1}^{s}\xi_{k}=1$. The weight vector of the attribute is expressed as $\omega ={\left(\omega_{1},\omega_{2},\ldots ,\omega_{m}\right)}^{T}$, $0\leq \omega_{j}\leq 1,j=1,2,\ldots ,m$, and $\sum_{j=1}^{m}\omega_{j}=1$. A certain scheme *M_i_* is rated by expert
$D_{k}$ according to attribute $C_{j}$, and gets an intuitionistic fuzzy decision matrix $R_{k}={\left(r_{ij}^{k}\right)}_{n\times m}$ where $r_{ij}^{k}=(\mu_{r_{ij}^{k}},\nu_{r_{ij}^{k}})$, $\mu_{r_{ij}^{k}}$, $\nu_{r_{ij}^{k}}$, $\pi_{r_{ij}^{k}}=1-\mu_{r_{ij}^{k}}-v_{r_{ij}^{k}}$ represents the satisfaction, dissatisfaction, and hesitation of expert $D_{k}$ for attribute $M_{i}$ under attribute $C_{j}$, respectively, k=1,2,…,s,i=1,2,…n,j=1,2,…m.

### 4.2. Intuitionistic Fuzzy Entropy Model for Decision Attribute Weight Determination

At present, in the field of intuitionistic fuzzy group decision making, scholars generally use entropy theory to determine the weight of decision attributes. Relative entropy determines the weight of the attribute by the relative entropy of the two attributes [63]. The calculation steps are as follows:

**Step 1:** Establishment of an intuitionistic fuzzy decision matrix.

The scores of each expert’s evaluation constitute an intuitionistic fuzzy decision matrix:(5)$$R_{k}=\left(\begin{array}{ccc} \left(\mu_{r_{11}^{k}},\nu_{r_{11}^{k}}\right) & \ldots & \left(\mu_{r_{1m}^{k}},\nu_{r_{1m}^{k}}\right)\\ \vdots & \ddots & \vdots \\ \left(\mu_{r_{n1}^{k}},\nu_{r_{n1}^{k}}\right) & \ldots & \left(\mu_{r_{nm}^{k}},\nu_{r_{nm}^{k}}\right) \end{array}\right)$$

**Step 2:** Take the i line from the above $R_{k}(k=1,2,\ldots ,s)$, forming a new matrix $B_{i}={\left(\left(\mu_{u_{ij}^{k}},v_{u_{ij}^{k}}\right)\right)}_{s\times m}(i=1,2,\ldots ,n,k=1,2,\ldots s)$ to indicate that the i scheme given by each expert conforms to the judgment matrix of m attributes.
(6)$$B_{i}=\left(\begin{array}{ccc} \left(\mu_{u_{i1}^{1}},\nu_{u_{i1}^{1}}\right) & \ldots & \left(\mu_{u_{im}^{1}},\nu_{u_{im}^{1}}\right)\\ \vdots & \ddots & \vdots \\ \left(\mu_{u_{i1}^{s}},\nu_{u_{i1}^{s}}\right) & \ldots & \left(\mu_{u_{im}^{s}},\nu_{u_{im}^{s}}\right) \end{array}\right)$$

**Step 3:** Use Formula (4) to find the entropy of each attribute based on Equation (7).
(7)$$H_{j}^{i}=-\frac{1}{s\ln2}\sum_{k=1}^{s}\mu_{r_{ij}^{k}}(x)\ln\mu_{r_{ij}^{k}}(x)+\nu_{r_{ij}^{k}}(x)\ln\nu_{r_{ij}^{k}}(x)-(1-\pi_{r_{ij}^{k}}(x)\times \ln((1-\pi_{r_{ij}^{k}}(x))-\pi_{r_{ij}^{k}}(x)\ln2)$$

**Step 4:** Obtain the entropy weight of each attribute.

According to the entropy theory, if the entropy value for each criterion is smaller across alternatives, it should provide decision-makers with the useful information. Therefore, the criterion should be assigned a bigger weight; otherwise, such a criterion will be judged unimportant by most decision-makers. In other words, such a criterion should be evaluated as a very small weight. If the information about weight $w_{j}^{i}$ of the criterion $C_{j}^{i}$ is completely unknown, the entropy weights for determining the criteria weight can be calculated as follows [69]:(8)$$w_{j}^{i}=\frac{1-H_{j}^{i}}{n-\sum_{j=1}^{m}H_{j}^{i}}$$

Further, obtain the objective weight of each attribute according to the judgment information under the *i* scheme $w_{j}^{i}=(\omega_{1}^{i},\omega_{2}^{i},\ldots ,\omega_{m}^{i})$. Then, the objective attribute weights of all the schemes can constitute a weight matrix, which is marked as:(9)$$\overset{\wedge }{w}=\left(\begin{array}{ccc} \omega_{1}^{1} & \cdots & \omega_{m}^{1}\\ \vdots & \ddots & \vdots \\ \omega_{1}^{n} & \cdots & \omega_{m}^{n} \end{array}\right)$$

**Step 5:** Find the optimal weight of each attribute.

Set the optimal weight of the attribute as $w=(\omega_{1},\omega_{2},\ldots \omega_{m})$; each row in the weight matrix can be thought of as the attribute weight probability distribution given by all decision-makers under each scenario. From the concept of relative entropy, the difference between the probability distribution *ω* and the optimal attribute weight should be small. Therefore, the following optimization model is constructed:(10)$$\begin{array}{l} \min RE(\omega )=\sum_{i=1}^{n}\sum_{j=1}^{m}\omega_{j}\mathrm{lg}(\frac{\omega_{j}}{\omega_{j}^{i}})\\ s.t\;\sum_{j=1}^{m}\omega_{j}=1,\;\omega_{j}>0\\ j=1,2,\ldots ,m \end{array}$$

Then, the optimal solution of the model is the most weight of the attribute $\omega^{*}=(\omega_{1}^{*},\omega_{2}^{*},\ldots ,\omega_{m}^{*})$, where:(11)$$\omega_{j}^{*}=\frac{\prod_{i=1}^{n}\omega_{j}^{i}}{\sum_{j=1}^{m}\prod_{i=1}^{n}\omega_{j}^{i}},j=1,2,\ldots m$$

### 4.3. Intuitionistic Fuzzy Comprehensive Entropy Model Based on Decision Expert Weight

In the process of group decision making, it is of great significance to objectively determine the weight of experts for more reliable decision making. The idea of cross entropy of intuitionistic fuzzy sets is that if the cross entropy between the two experts is smaller—namely, if the difference between their scores is smaller—then the individual evaluation results are relatively good, and a larger weight is given; while in contrast, a smaller weight is given. This paper proposed a method to obtain expert weight by combining cross entropy and entropy. The specific steps are as follows:

**Step 6:** According to the attribute weight value calculated in step 5, combined with the original expert weight value, the IHWA in Definition 6 is used to aggregate the scheme information, and the expert $D_{k}$ evaluation result $y_{ki}$ for scheme $M_{i}$, as well as individual and the expert group evaluation result $x_{i}$ are obtained.
$$\begin{array}{c} y_{ki}=\overset{m}{\underset{j=1}{⊕}}r_{ij}^{k}\omega_{j}=(1-\prod_{j=1}^{m}{\left(1-\mu_{r_{ij}^{k}}\right)}^{\omega_{j}},\prod_{j=1}^{m}{\left(\nu_{r_{ij}^{k}}\right)}^{\omega_{j}}),k=1,2,\ldots ,s;i=1,2,\ldots ,n\\ x_{i}=\overset{s}{\underset{k=1}{⊕}}y_{ki}\xi_{k}=(1-\prod_{k=1}^{s}{\left[1-(1-\prod_{j=1}^{m}{\left(1-\mu_{r_{ij}^{k}}\right)}^{\omega_{j}})\right]}^{\xi_{k}},\prod_{k=1}^{s}{\left[\prod_{j=1}^{m}{\left(\nu_{r_{ij}^{k}}\right)}^{\omega_{j}}\right]}^{\xi_{k}}),i=1,2,\ldots ,n \end{array}$$

**Step 7:** According to the cross-entropy formula in reference [26,62], the cross entropy between the individual and group scoring results can be obtained from the individual evaluation result vector $Y_{k}={\left(y_{k1},y_{k2},\ldots ,y_{kn}\right)}^{T}$ and the group evaluation result vector $X={\left(x_{1},x_{2},\ldots ,x_{n}\right)}^{T}$.
(12)$$D(Y_{k},X)=\sum_{i=1}^{n}\left[\mu_{ki}\ln\frac{\mu_{ki}}{\frac{1}{2}\left(\mu_{ki}+\mu_{i}\right)}+v_{ki}\ln\frac{v_{ki}}{\frac{1}{2}\left(v_{ki}+v_{i}\right)}\right]+\sum_{i=1}^{n}\left[\mu_{i}\ln\frac{\mu_{i}}{\frac{1}{2}\left(\mu_{ki}+\mu_{i}\right)}+v_{i}\ln\frac{v_{i}}{\frac{1}{2}\left(v_{ki}+v_{i}\right)}\right]$$
where $\mu_{ki}=1-\prod_{j=1}^{m}{\left(1-\mu_{r_{ij}^{k}}\right)}^{\omega_{j}}$, $v_{ki}=\prod_{j=1}^{m}{\left(v_{r_{ij}^{k}}\right)}^{\omega_{j}}$, $\mu_{i}=1-\prod_{k=1}^{s}{\left(1-\mu_{ki}\right)}^{\xi_{k}}$, $v_{i}=\prod_{k=1}^{s}{\left(v_{ki}\right)}^{\xi_{k}}$, i=1,2,…,n.

Furthermore, the weight of experts based on cross entropy is obtained as follows:(13)$$r_{k}=\frac{1}{D(Y_{k},X)}/\sum_{k=1}^{s}\frac{1}{D(Y_{k},X)}$$

Then, $0\leq r_{k}\leq 1;k=1,2,\ldots ,s;\sum_{k=1}^{s}r_{k}=1$.

**Step 8:** Then, the entropy value of the evaluation value $r_{k}$ be $E_{k}$. $E_{k}$ can be computed by the following equation [70]:(14)$$E_{k}=\frac{1}{n}\sum_{i=1}^{n}\frac{\min\left\{\mu_{ki},v_{ki}\right\}+\pi_{ki}}{\max\left\{\mu_{ki},v_{ki}\right\}+\pi_{ki}}$$

Furthermore, the weight of the expert based on entropy is $e_{k}=(1-E_{k})/(s-\sum_{k=1}^{s}E_{k})$, where: $0\leq e_{k}\leq 1;k=1,2,\ldots ,s;\sum_{k=1}^{s}e_{k}=1$.

**Step 9:** From the weight $r_{k}$ of cross entropy and the weight $e_{k}$ based on entropy, the expert weight is obtained by combining the weighting method:(15)$$\gamma_{k}=\alpha r_{k}+\beta e_{k},k=1,2,\ldots ,s$$
where 0≤α≤1,α+β=1.

### 4.4. Overall Scheme Sorting Model Based on IFHA and IHWA Operators

**Step 10:** The intuitionistic fuzzy hybrid average operator (IFHA) is used to aggregate the fuzzy evaluation value of each expert under each scheme.
(16)$$IFHA_{\gamma_{k},w}(r_{ij}^{1},r_{ij}^{2},\cdots ,r_{ij}^{s})=\left(1-\prod_{k=1}^{s}{\left(1-\mu_{{\dot{r}}_{ij}^{\sigma (k)}}\right)}^{w_{k}},\prod_{k=1}^{s}{\left(\nu_{{}_{{\dot{r}}_{ij}^{\sigma (k)}}}\right)}^{w_{k}}\right)$$

The fuzzy decision matrix between groups is obtained: $\ddot{R}={\left({\ddot{r}}_{ij}\right)}_{n\times m}$.

**Step 11:** The comprehensive attribute values of each scheme are obtained by using the intuitionistic fuzzy weighted average operator (IFWA) by Definition 6.
(17)$$IFWA_{\omega }({\ddot{r}}_{ij},{\ddot{r}}_{ij},\ldots ,{\ddot{r}}_{ij})=\left(1-\prod_{j=1}^{m}{\left(1-\mu_{{\ddot{r}}_{ij}}\right)}^{\omega_{j}},\prod_{j=1}^{m}{\left(\nu_{{\ddot{r}}_{ij}}\right)}^{\omega_{j}}\right)$$

**Step 12:** The score of the comprehensive attribute value is $s({\ddot{r}}_{i}),i=1,2,\ldots ,n$, from Definition 3, and the final ranking of the scheme is obtained.

## 5. Case Study Analysis

### 5.1. Background Description

In a large hydropower station project in China, the hydropower station adopts hybrid development. The normal storage water level of the reservoir is 398 m, the storage capacity is 63.3 million m^3^, the design reference flow is 2640.9 m^3^/s, the installed capacity of the power station is 772 MW, and the average annual power generation is 3.303 billion KWH. The main engineering quantities include (excluding temporary and diversion projects): earthwork excavation 16.1577 million m^3^, stone excavation 2.1824 million m^3^, concrete pouring 2.1112 million m^3^, earthwork filling 3.2906 million m^3^, masonry project 455,500 m^3^, steel bar 41,700 t, curtain grouting 13,300 m, consolidation grouting 2.63 m, concrete impervious wall 100,200 m^2^, metal structure installation 10,700 t, install 19 hoist sets.

The scale of the hydropower station project is relatively large and the geological condition is not very good. The owner intended to adopt a project delivery mode; alternative ones include the DBB mode, DB mode, and EPC mode. The decision index was the 15 key influencing factors index of the index system established in Section 2. The owner unit engaged five senior experts in the relevant fields, namely, the owner representative personnel *D*_1_, the construction technology expert *D*_2_, the cost engineer *D*_3_, economic experts *D*_4_, and environmental experts *D*_5_.

### 5.2. Determination of Attribute Weight

According to the above description, the specific decision system can be expressed as follows: the set of decision makers composed of five experts is $D=\{D_{1},D_{2},\ldots ,D_{5}\}$; on the basis of scientific research, the weight vector given to the five experts in advance is $\xi {=(0.2,0.1,0.22,0.17,0.31)}^{T}$ (only a preliminary assumption that the weight will vary according to different items). The scheme set of the three alternative trading modes is $M=\{M_{1},M_{2},M_{3}\}$. The attribute set of 15 evaluation indexes established according to Table 1 is $C=\{C_{1},C_{2},\ldots ,C_{15}\}$. Expert $D_{k}$ evaluates scheme $M_{i}$ according to attribute $C_{j}$, and obtains the fuzzy decision matrix $R^{k}={\left(r_{ij}^{k}\right)}_{5\times 13}(k=1,2\ldots 5;i=1,2,3;j=1,2,\ldots ,15)$. The specific evaluation process for the project is as follows:

(1) Establishment of intuitionistic fuzzy matrix. Based on the rating of the three alternative models by five experts, the intuitionistic fuzzy matrix is as follows:$$R_{1}=\left(\begin{array}{cccccccccccccccc} & C_{1} & C_{2} & C_{3} & C_{4} & C_{5} & C_{6} & C_{7} & C_{8} & C_{9} & C_{10} & C_{11} & C_{12} & C_{13} & C_{14} & C_{15}\\ M_{1} & \left(0.6,0.2\right) & \left(0.5,0.1\right) & \left(0.6,0.3\right) & \left(0.7,0.1\right) & \left(0.8,0.2\right) & \left(0.6,0.3\right) & \left(0.3,0.1\right) & \left(0.7,0.1\right) & \left(0.6,0.1\right) & \left(0.6,0.2\right) & \left(0.5,0.5\right) & \left(0.6,0.1\right) & \left(0.7,0.1\right) & \left(0.6,0.2\right) & \left(0.4,0.2\right)\\ M_{2} & \left(0.8,0.1\right) & \left(0.7,0.2\right) & \left(0.6,0.2\right) & \left(0.5,0.4\right) & \left(0.7,0.3\right) & \left(0.6,0.3\right) & \left(0.8,0.1\right) & \left(0.8,0.2\right) & \left(0.7,0.1\right) & \left(0.6,0.4\right) & \left(0.4,0.6\right) & \left(0.6,0.3\right) & \left(0.7,0.2\right) & \left(0.6,0.3\right) & \left(0.5,0.4\right)\\ M_{3} & \left(0.2,0.7\right) & \left(0.3,0.5\right) & \left(0.5,0.2\right) & \left(0.4,0.3\right) & \left(0.2,0.5\right) & \left(0.6,0.2\right) & \left(0.7,0.2\right) & \left(0.3,0.5\right) & \left(0.5,0.2\right) & \left(0.6,0.1\right) & \left(0.6,0.1\right) & \left(0.3,0.2\right) & \left(0.4,0.4\right) & \left(0.3,0.5\right) & \left(0.5,0.2\right) \end{array}\right)$$
$$R_{2}=\left(\begin{array}{cccccccccccccccc} & C_{1} & C_{2} & C_{3} & C_{4} & C_{5} & C_{6} & C_{7} & C_{8} & C_{9} & C_{10} & C_{11} & C_{12} & C_{13} & C_{14} & C_{15}\\ M_{1} & \left(0.5,0.2\right) & \left(0.6,0.1\right) & \left(0.5,0.4\right) & \left(0.6,0.3\right) & \left(0.8,0.1\right) & \left(0.7,0.2\right) & \left(0.4,0.1\right) & \left(0.6,0.3\right) & \left(0.7,0.1\right) & \left(0.7,0.1\right) & \left(0.6,0.2\right) & \left(0.5,0.4\right) & \left(0.8,0.1\right) & \left(0.7,0.2\right) & \left(0.5,0.4\right)\\ M_{2} & \left(0.6,0.3\right) & \left(0.6,0.1\right) & \left(0.7,0.2\right) & \left(0.4,0.5\right) & \left(0.3,0.6\right) & \left(0.4,0.5\right) & \left(0.2,0.7\right) & \left(0.6,0.4\right) & \left(0.2,0.6\right) & \left(0.7,0.2\right) & \left(0.4,0.5\right) & \left(0.7,0.2\right) & \left(0.6,0.3\right) & \left(0.2,0.6\right) & \left(0.4,0.5\right)\\ M_{3} & \left(0.3,0.2\right) & \left(0.4,0.4\right) & \left(0.6,0.2\right) & \left(0.3,0.6\right) & \left(0.5,0.1\right) & \left(0.6,0.2\right) & \left(0.6,0.2\right) & \left(0.5,0.4\right) & \left(0.7,0.2\right) & \left(0.4,0.1\right) & \left(0.5,0.4\right) & \left(0.6,0.1\right) & \left(0.3,0.4\right) & \left(0.7,0.1\right) & \left(0.6,0.2\right) \end{array}\right)$$
$$R_{3}=\left(\begin{array}{cccccccccccccccc} & C_{1} & C_{2} & C_{3} & C_{4} & C_{5} & C_{6} & C_{7} & C_{8} & C_{9} & C_{10} & C_{11} & C_{12} & C_{13} & C_{14} & C_{15}\\ M_{1} & \left(0.4,0.3\right) & \left(0.5,0.3\right) & \left(0.6,0.4\right) & \left(0.2,0.7\right) & \left(0.5,0.4\right) & \left(0.6,0.2\right) & \left(0.6,0.1\right) & \left(0.2,0.5\right) & \left(0.4,0.1\right) & \left(0.5,0.3\right) & \left(0.2,0.7\right) & \left(0.8,0.1\right) & \left(0.7,0.2\right) & \left(0.5,0.2\right) & \left(0.7,0.2\right)\\ M_{2} & \left(0.7,0.1\right) & \left(0.6,0.2\right) & \left(0.3,0.2\right) & \left(0.5,0.5\right) & \left(0.6,0.1\right) & \left(0.7,0.3\right) & \left(0.4,0.1\right) & \left(0.5,0.4\right) & \left(0.3,0.4\right) & \left(0.3,0.5\right) & \left(0.1,0.7\right) & \left(0.6,0.2\right) & \left(0.7,0.1\right) & \left(0.3,0.5\right) & \left(0.4,0.1\right)\\ M_{3} & \left(0.2,0.5\right) & \left(0.3,0.2\right) & \left(0.5,0.4\right) & \left(0.2,0.7\right) & \left(0.5,0.3\right) & \left(0.6,0.1\right) & \left(0.7,0.2\right) & \left(0.4,0.5\right) & \left(0.6,0.3\right) & \left(0.5,0.2\right) & \left(0.6,0.2\right) & \left(0.7,0.1\right) & \left(0.2,0.3\right) & \left(0.5,0.3\right) & \left(0.6,0.1\right) \end{array}\right)$$
$$R_{4}=\left(\begin{array}{cccccccccccccccc} & C_{1} & C_{2} & C_{3} & C_{4} & C_{5} & C_{6} & C_{7} & C_{8} & C_{9} & C_{10} & C_{11} & C_{12} & C_{13} & C_{14} & C_{15}\\ M_{1} & \left(0.5,0.2\right) & \left(0.4,0.1\right) & \left(0.2,0.6\right) & \left(0.3,0.5\right) & \left(0.6,0.1\right) & \left(0.4,0.1\right) & \left(0.5,0.2\right) & \left(0.3,0.4\right) & \left(0.5,0.2\right) & \left(0.2,0.6\right) & \left(0.3,0.1\right) & \left(0.7,0.3\right) & \left(0.3,0.3\right) & \left(0.7,0.1\right) & \left(0.5,0.4\right)\\ M_{2} & \left(0.6,0.2\right) & \left(0.5,0.4\right) & \left(0.4,0.5\right) & \left(0.4,0.5\right) & \left(0.5,0.2\right) & \left(0.6,0.1\right) & \left(0.5,0.4\right) & \left(0.2,0.6\right) & \left(0.5,0.2\right) & \left(0.6,0.1\right) & \left(0.5,0.4\right) & \left(0.3,0.1\right) & \left(0.5,0.3\right) & \left(0.3,0.4\right) & \left(0.3,0.4\right)\\ M_{3} & \left(0.2,0.6\right) & \left(0.5,0.3\right) & \left(0.4,0.2\right) & \left(0.2,0.7\right) & \left(0.5,0.3\right) & \left(0.6,0.1\right) & \left(0.7,0.2\right) & \left(0.4,0.5\right) & \left(0.6,0.3\right) & \left(0.5,0.2\right) & \left(0.6,0.2\right) & \left(0.7,0.1\right) & \left(0.2,0.3\right) & \left(0.6,0.2\right) & \left(0.6,0.1\right) \end{array}\right)$$
$$R_{5}=\left(\begin{array}{cccccccccccccccc} & C_{1} & C_{2} & C_{3} & C_{4} & C_{5} & C_{6} & C_{7} & C_{8} & C_{9} & C_{10} & C_{11} & C_{12} & C_{13} & C_{14} & C_{15}\\ M_{1} & \left(0.5,0.1\right) & \left(0.4,0.2\right) & \left(0.5,0.4\right) & \left(0.3,0.5\right) & \left(0.6,0.1\right) & \left(0.7,0.3\right) & \left(0.5,0.1\right) & \left(0.3,0.1\right) & \left(0.5,0.3\right) & \left(0.3,0.6\right) & \left(0.2,0.7\right) & \left(0.7,0.1\right) & \left(0.8,0.1\right) & \left(0.4,0.5\right) & \left(0.7,0.1\right)\\ M_{2} & \left(0.4,0.1\right) & \left(0.7,0.1\right) & \left(0.4,0.2\right) & \left(0.6,0.1\right) & \left(0.2,0.1\right) & \left(0.5,0.4\right) & \left(0.5,0.2\right) & \left(0.3,0.2\right) & \left(0.3,0.5\right) & \left(0.4,0.4\right) & \left(0.2,0.5\right) & \left(0.7,0.2\right) & \left(0.6,0.3\right) & \left(0.5,0.2\right) & \left(0.5,0.3\right)\\ M_{3} & \left(0.3,0.4\right) & \left(0.2,0.5\right) & \left(0.4,0.5\right) & \left(0.8,0.1\right) & \left(0.6,0.3\right) & \left(0.4,0.1\right) & \left(0.5,0.2\right) & \left(0.5,0.1\right) & \left(0.7,0.1\right) & \left(0.3,0.4\right) & \left(0.4,0.2\right) & \left(0.6,0.2\right) & \left(0.3,0.1\right) & \left(0.6,0.1\right) & \left(0.6,0.2\right) \end{array}\right)$$

(2) The X line of the above five intuitionistic fuzzy matrices is taken out, and the new matrix $B_{i}={\left(\left(\mu_{u_{ij}^{k}},v_{u_{ij}^{k}}\right)\right)}_{5\times 15}(i=1,2,3,k=1,2,\ldots 5)$ represents the judgment matrix of 15 attributes of the first scheme by five experts, respectively. For example, take out the first line and construct a new matrix $B_{1}={\left(\left(\mu_{u_{1j}^{k}},v_{u_{1j}^{k}}\right)\right)}_{5\times 15}$ that represents the judgment matrix of five experts on the 15 attributes of the first scenario, which are shown as follows:$$B_{1}=\left(\begin{array}{cccccccccccccccc} & C_{1} & C_{2} & C_{3} & C_{4} & C_{5} & C_{6} & C_{7} & C_{8} & C_{9} & C_{10} & C_{11} & C_{12} & C_{13} & C_{14} & C_{15}\\ R_{1} & \left(0.6,0.2\right) & \left(0.5,0.1\right) & \left(0.6,0.3\right) & \left(0.7,0.1\right) & \left(0.8,0.2\right) & \left(0.6,0.3\right) & \left(0.3,0.1\right) & \left(0.7,0.1\right) & \left(0.6,0.1\right) & \left(0.6,0.2\right) & \left(0.5,0.5\right) & \left(0.6,0.1\right) & \left(0.7,0.1\right) & \left(0.6,0.2\right) & \left(0.4,0.2\right)\\ R_{2} & \left(0.5,0.2\right) & \left(0.6,0.1\right) & \left(0.5,0.4\right) & \left(0.6,0.3\right) & \left(0.8,0.1\right) & \left(0.7,0.2\right) & \left(0.4,0.1\right) & \left(0.6,0.3\right) & \left(0.7,0.1\right) & \left(0.7,0.1\right) & \left(0.6,0.2\right) & \left(0.5,0.4\right) & \left(0.8,0.1\right) & \left(0.7,0.2\right) & \left(0.5,0.4\right)\\ R_{3} & \left(0.4,0.3\right) & \left(0.5,0.3\right) & \left(0.6,0.4\right) & \left(0.2,0.7\right) & \left(0.5,0.4\right) & \left(0.6,0.2\right) & \left(0.6,0.1\right) & \left(0.2,0.5\right) & \left(0.4,0.1\right) & \left(0.5,0.3\right) & \left(0.2,0.7\right) & \left(0.8,0.1\right) & \left(0.7,0.2\right) & \left(0.5,0.2\right) & \left(0.7,0.2\right)\\ R_{4} & \left(0.5,0.2\right) & \left(0.4,0.1\right) & \left(0.2,0.6\right) & \left(0.3,0.5\right) & \left(0.6,0.1\right) & \left(0.4,0.1\right) & \left(0.5,0.2\right) & \left(0.3,0.4\right) & \left(0.5,0.2\right) & \left(0.2,0.6\right) & \left(0.3,0.1\right) & \left(0.7,0.3\right) & \left(0.3,0.3\right) & \left(0.7,0.1\right) & \left(0.5,0.4\right)\\ R_{5} & \left(0.5,0.1\right) & \left(0.4,0.2\right) & \left(0.5,0.4\right) & \left(0.3,0.5\right) & \left(0.6,0.1\right) & \left(0.7,0.3\right) & \left(0.5,0.1\right) & \left(0.3,0.1\right) & \left(0.5,0.3\right) & \left(0.3,0.6\right) & \left(0.2,0.7\right) & \left(0.7,0.1\right) & \left(0.8,0.1\right) & \left(0.4,0.5\right) & \left(0.7,0.1\right) \end{array}\right)$$

(3) The entropies of 15 attributes are calculated, and the results are as shown in Table 2.

(4) According to Formula (8), the entropy weight of each attribute of the information given by the expert is calculated, and the results are as follows:

$\omega_{1}^{1}=0.052$, $\omega_{2}^{1}=0.063$, $\omega_{3}^{1}=0.023$, $\omega_{4}^{1}=0.061$, $\omega_{5}^{1}=0.114$, $\omega_{6}^{1}=0.065$, $\omega_{7}^{1}=0.077$, $\omega_{8}^{1}=0.058$, $\omega_{9}^{1}=0.088$, $\omega_{10}^{1}=0.076$, $\omega_{11}^{1}=0.065$, $\omega_{12}^{1}=0.124$, $\omega_{13}^{1}=0.158$, $\omega_{14}^{1}=0.079$, $\omega_{15}^{1}=0.084$;

$\omega_{1}^{2}=0.136$, $\omega_{2}^{2}=0.129$, $\omega_{3}^{2}=0.061$, $\omega_{4}^{2}=0.044$, $\omega_{5}^{2}=0.078$, $\omega_{6}^{2}=0.066$, $\omega_{7}^{2}=0.113$, $\omega_{8}^{2}=0.060$, $\omega_{9}^{2}=0.085$, $\omega_{10}^{2}=0.071$, $\omega_{11}^{2}=0.064$, $\omega_{12}^{2}=0.086$, $\omega_{13}^{2}=0.086$, $\omega_{14}^{2}=0.089$, $\omega_{15}^{2}=0.104$;

$\omega_{1}^{3}=0.057$, $\omega_{2}^{3}=0.019$, $\omega_{3}^{3}=0.030$, $\omega_{4}^{3}=0.082$, $\omega_{5}^{3}=0.040$, $\omega_{6}^{3}=0.080$, $\omega_{7}^{3}=0.067$, $\omega_{8}^{3}=0.021$, $\omega_{9}^{3}=0.061$, $\omega_{10}^{3}=0.040$, $\omega_{11}^{3}=0.082$, $\omega_{12}^{3}=0.083$, $\omega_{13}^{3}=0.069$, $\omega_{14}^{3}=0.094$, $\omega_{15}^{3}=0.063$.

### 5.3. Determination of Expert Weight

(1) According to the formula, the individual evaluation result $Y_{k}=(y_{k1},y_{k2},y_{k3},y_{k4},y_{k5})$ of expert *D_k_* can be calculated, and the results are shown as follows:$$y_{ki}=\left(\begin{array}{cccccc} & D_{1} & D_{2} & D_{3} & D_{4} & D_{5}\\ M_{1} & \left(0.6435,0.1416\right) & \left(0.6375,0.2118\right) & \left(0.6584,0.1953\right) & \left(0.5568,0.2245\right) & \left(0.6311,0.14874\right)\\ M_{2} & \left(0.6404,0.2726\right) & \left(0.5705,0.3169\right) & \left(0.5741,0.1946\right) & \left(0.4367,0.1704\right) & \left(0.5538,0.2306\right)\\ M_{3} & \left(0.3928,0.2496\right) & \left(0.6527,0.1901\right) & \left(0.5775,0.1808\right) & \left(0.5773,0.1797\right) & \left(0.5424,0.1934\right) \end{array}\right)$$

The results of the expert group score can be obtained as follows:X=((0.6290,0.1236),(0.5616,0.2167),(0.5434,0.1977),(0.7091,0.1655),(0.5803,0.2049))

(2) According to Formula (9), the weight of experts based on cross entropy is obtained as follows:$$r_{k}=(0.1368,0.0512,0.1391,0.1459,0.5270)$$

(3) According to Formula (10), the weight of experts based on entropy is calculated as follows:$$e_{k}=(0.1994,0.2017,0.2024,0.1986)$$

(4) Setting α=0.6 and β=0.4, the final weight of experts $\gamma_{k}=(0.1618,0.1114,0.1644,0.1667,0.3957)$ is calculated by the combined weight method.

### 5.4. Ranking of Overall Schemes and Patterns Comparison

(1) Information aggregation of the fuzzy evaluation value of all the experts under each program is carried out according to Formula (13), and the results are as follows:$$R=\left(\begin{array}{cccccccccccccccc} & C_{1} & C_{2} & C_{3} & C_{4} & C_{5} & C_{6} & C_{7} & C_{8} & C_{9} & C_{10} & C_{11} & C_{12} & C_{13} & C_{14} & C_{15}\\ M_{1} & \left(0.475,0.211\right) & \left(0.427,0.166\right) & \left(0.489,0.437\right) & \left(0.406,0.407\right) & \left(0.637,0.167\right) & \left(0.558,0.222\right) & \left(0.455,0.144\right) & \left(0.425,0.251\right) & \left(0.501,0.186\right) & \left(0.458,0.355\right) & \left(0.356,0.447\right) & \left(0.659,0.162\right) & \left(0.677,0.172\right) & \left(0.460,0.563\right) & \left(0.439,0.543\right)\\ M_{2} & \left(0.617,0.147\right) & \left(0.597,0.201\right) & \left(0.457,0.271\right) & \left(0.473,0.357\right) & \left(0.478,0.148\right) & \left(0.570,0.294\right) & \left(0.519,0.206\right) & \left(0.502,0.348\right) & \left(0.415,0.315\right) & \left(0.505,0.313\right) & \left(0.322,0.560\right) & \left(0.557,0.222\right) & \left(0.605,0.242\right) & \left(0.396,0.342\right) & \left(0.592,0.477\right)\\ M_{3} & \left(0.217,0.488\right) & \left(0.311,0.372\right) & \left(0.427,0.311\right) & \left(0.436,0.400\right) & \left(0.464,0.312\right) & \left(0.530,0.143\right) & \left(0.624,0.228\right) & \left(0.396,0.370\right) & \left(0.586,0.235\right) & \left(0.460,0.244\right) & \left(0.563,0.193\right) & \left(0.607,0.174\right) & \left(0.381,0.277\right) & \left(0.443,0.236\right) & \left(0.372,0.246\right) \end{array}\right)$$

(2) According to Formula (14), the comprehensive attribute values of each scheme are calculated:$${\ddot{r}}_{1}=(0.5730,0.2075),\;\;{\ddot{r}}_{2}=(0.5318,0.2216),\;\;{\ddot{r}}_{3}=(0.4904,0.2579).$$

(3) According to Definition 3, the score of the comprehensive attribute value is calculated and sorted.
$$s({\ddot{r}}_{1})=0.2798,\;\;s({\ddot{r}}_{2})=0.3215,\;\;\;s({\ddot{r}}_{3})=0.2801.$$


Furthermore, the final order of the scheme is obtained: A>B>C. Therefore, the second transaction mode of DBB is the optimal choice.

## 6. Conclusions

The PDS determines the project performance and is critical to water engineering project success. For a given water engineering project, selecting the proper PDS is one of the decisive factors. PDS selection is a typical multi-attribute decision-making problem that can be effectively solved by group decision making. IFS is always used to solve complex decision-making problems, especially multi-attribute group decision-making problems, under uncertain circumstances. Based on the IFS group decision-making model and intuitionistic fuzzy entropy, a new decision-making support method for PDS selection was proposed. In order to reduce the loss of judgment information and improve the objectivity and fairness of group decision making, two operators—IFHA and IFWA on PDS decision—were addressed to calculate the final decision weights. The case study showed that the method proposed in this paper is an effective approach and has potential practical application that would help water engineering project owners in PDS selection. The model proposed in this paper can also be applied to solve similar decision-making problems. The method proposed in this paper is an effective group decision method and can help the water engineering project owners in PDS selection, but the decision making that is influenced by expert subjective evaluation has to be considered, and how to reduce the influence of the subjective evaluations from experts is the next research direction.

## Figures and Tables

**Table 1 entropy-21-01101-t001:** Index system of influencing factors for a water engineering project delivery system (PDS).

Level I Indexes	Level II Indexes	Meanings of Level II Indexes	Source
Owner’s Characteristics	Owner liability	The employer’s expectation of liability for as few of the participants as possible	[7,11,13,17,18,24,27,28,29,30,31,32,33,34,35,36]
	Owner participation	Willingness and degree of participation of owners during the whole life cycle of the project	[10,11,17,28,30,31,34]
	Owner ability	Owner’s own ability, such as decision-making ability, project control and organization ability, and project management ability	[13,17,28,31,34,37,38,39]
	Risk allocation	Expected commitment of owners to risks and losses (that is, whether it is shared equally with the contractor, or whether the owner bears most of the risk, or the contractor bears the majority of the risk)	[12,13,21,28,34]
	Owner design control	The willingness and degree of the owner to participate in the design	[11,31,32,34,38,40,41]
Project characteristics	Project scale	Compared with the average scale of the engineering project in the industry	[7,12,13,17,18,28,30,31,32,41,42,43,44,45,46,47,48,49]
	Project complexity	Whether the project needs a breakthrough in construction methods, technology and management, the complexity of technology, the uncertainty of the project, the observability of the characteristic values of engineering products, and so on	[8,10,11,12,13,17,18,24,27,31,32,33,39,40,41,42,43,44,48,49,50,51,52,53,54]
	Project type	What types of projects (e.g., housing construction projects, infrastructure projects, industrial projects, etc.)	[7,10,15,17,18,30,31,44,47,48,55,56,57]
	Project scope clarity	Clarity of project scoping	[8,10,11,18,21,31,38,40,41,42,43,44,45,48,50,51,58]
	Project flexibility	Flexibility of expected design and construction changes in the implementation of the project	[8,10,11,13,17,18,21,24,27,28,29,31,32,33,39,44,52,54]
	Project disputes	The severity of potential disputes in the course of project construction (e.g., serious disputes, etc.)	[7,17,18,24,27,28,31,32,41,44,49,53,59]
External environment	Market competition	Competition level in the contractor market	[8,11,12,15,18,21,29,30,31,49,52]
	Accessibility of materials	The extent to which the necessary raw materials for the project are difficult to purchase in the market	[8,10,15,21,30,38]
	Availability of technology	The degree of difficulty in obtaining the necessary technology for project construction in the market	[8,10,15,17,18,21,30,31,49]
	Impact of laws and regulations	The limitation of the perfection of laws and regulations on the PDS	[8,11,12,15,17,18,21,30,32,48,49,53]

**Table 2 entropy-21-01101-t002:** The entropies of the 15 attributes for the three alternatives.

Alternatives	Indicator Entropy Value
M_1_	$H_{1}^{1}=0.8874$, $H_{2}^{1}=0.8560$, $H_{3}^{1}=0.9461$, $H_{4}^{1}=0.8552$, $H_{5}^{1}=0.7390$, $H_{6}^{1}=0.8611$, $H_{7}^{1}=0.8388$, $H_{8}^{1}=0.8759$, $H_{9}^{1}=0.8155$, $H_{10}^{1}=0.8446$, $H_{11}^{1}=0.8698$, $H_{12}^{1}=0.7550$, $H_{13}^{1}=0.7057$, $H_{14}^{1}=0.7957$, $H_{15}^{1}=0.8257$
M_2_	$H_{1}^{2}=0.7648$, $H_{2}^{2}=0.7956$, $H_{3}^{2}=0.9130$, $H_{4}^{2}=0.930$, $H_{5}^{2}=0.8803$, $H_{6}^{2}=0.9012$, $H_{7}^{2}=0.8196$, $H_{8}^{2}=0.9039$, $H_{9}^{2}=0.8683$, $H_{10}^{2}=0.8873$, $H_{11}^{2}=0.8988$, $H_{12}^{2}=0.8551$, $H_{13}^{2}=0.8478$, $H_{14}^{2}=0.8357$, $H_{15}^{1}=0.8942$
M_3_	$H_{1}^{3}=0.9032$, $H_{2}^{3}=0.9634$, $H_{3}^{3}=0.9376$, $H_{4}^{3}=0.8089$, $H_{5}^{3}=0.9096$, $H_{6}^{3}=0.7975$, $H_{7}^{3}=0.8233$, $H_{8}^{3}=0.9459$, $H_{9}^{3}=0.8360$, $H_{10}^{3}=0.8966$, $H_{11}^{3}=0.7832$, $H_{12}^{3}=0.7784$, $H_{13}^{3}=0.9235$, $H_{14}^{3}=0.8477$, $H_{15}^{3}=0.8934$

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
