# Peer review of "Intuitionistic Fuzzy Entropy for Group Decision Making of Water Engineering Project Delivery System Selection"

_entropy, 2019, doi:10.3390/e21111101_

Round 1

Reviewer 1 Report

From mathematical point of view, the paper is correct, but I cannot comment it from water engineering point of view. I have only one critical remark: the first three operations from Definition are given in the first Atanassov’s paper on intuitionistic fuzzy sets and the second two – in

Kumar De, R. Biswas, A. R. Roy, Some operations on intuitionistic fuzzy sets, Fuzzy Sets and Systems, Vol. 114 (2000), No. 4, 477-484

(see, also

Atanassov, K. On Intuitionistic Fuzzy Sets Theory, Springer, Berlin, 2012).

Author Response

Response to the Editor and Reviewers’ Comments

Manuscript ID: entropy-628521

Title: "Intuitionistic Fuzzy Entropy for Group Decision Making of Water Engineering Project Delivery System Selection"

Author(s): XUN LIU *, FEI QIAN, LINGNA LIN, KUN ZHANG, LIANBO ZHU

Revision due before: 10-Nov-2019

We appreciate the time and effort of the Reviewers and the Editor in reviewing our manuscript. The reviews are very helpful for us to improve the manuscript. Because of the comments from both the Editor and the Reviewers, we have made significant changes and have rewritten parts of the manuscript. Point to point responses to all comments are as follows.

Reviewer: 1

General Comment 1:

From mathematical point of view, the paper is correct, but I cannot comment it from water engineering point of view. I have only one critical remark: the first three operations from Definition are given in the first Atanassov’s paper on intuitionistic fuzzy sets and the second two – in Kumar De, R. Biswas, A. R. Roy, Some operations on intuitionistic fuzzy sets, Fuzzy Sets and Systems, Vol. 114 (2000), No. 4, 477-484

(see, also Atanassov, K. On Intuitionistic Fuzzy Sets Theory, Springer, Berlin, 2012).

Response:

We thank for reviewer’s comment. We have incorporated the relevant references in our manuscript. Please see line 126-135.

References were also incorporated in manuscript:

    [61] Atanassov, K.T., Intuitionistic fuzzy sets. Fuzzy Sets and Systems, 1986. 20(1): p. 87-96.

    [64] De, S.K., R. Biswas, and A.R. Roy, Some operations on intuitionistic fuzzy sets. Fuzzy sets and systems, 2000. 114(3): p. 477-484.

    [66] Xu, Z., Intuitionistic fuzzy aggregation operators. Fuzzy Systems, IEEE Transactions on, 2007. 15(6): p. 1179-1187.

Reviewer 2 Report

The paper " Intuitionistic Fuzzy Entropy for Group Decision Making of Water Engineering Project Delivery
System Selection" addresses the problem of Group DEcision Making through a mthodology based on the
Intuitionistic Fuzzy Entropy approach for the assessment of the weight of ingluencing factor of transaction modes
and the the weight of decision experts in a Delivery System Selection.
Nevertheless it has the following problems:

Forms (Typo): The paper can´t published in its current state due to differents kind of mistakes ranging from
i) Typo (v.gr lines 17,18,117,156,170..),

ii) Loosen acronyms (v.gr lines 61,62)

to iii)complete loosen paragraphs or leading to misunderstandings
(v.gr. lines 88-100,109-112,128-129,152-154,194-197,210-215).

Technical Contribution:

iv) It is quite confusing in lines 194-197 how weight entropies of each attribute can provide
a sound decision-making assessment when an analysis of suitable thresholds has´not been formally
achieved. In my opinion, an interval preference relation is enough for decision
makers to provide preference relation.

v). It seems tha the proposed method is a simple rewrite of previous work.
There are not too many actual contributions and novelties.

vi). The comparison section is incomplete. The authors are suggested to make comparisons with the state-of-the-art
improving methods. It is difficult to evaluate the advantages of the proposed method.

vii) One could expect the contribution to be based on the PDS, however, the results obtained are
not contrasted against other methodologies.

Author Response

Response to the Editor and Reviewers’ Comments

Manuscript ID: entropy-628521

Title: "Intuitionistic Fuzzy Entropy for Group Decision Making of Water Engineering Project Delivery System Selection"

Author(s): XUN LIU *, FEI QIAN, LINGNA LIN, KUN ZHANG, LIANBO ZHU

Revision due before: 10-Nov-2019

We appreciate the time and effort of the Reviewers and the Editor in reviewing our manuscript. The reviews are very helpful for us to improve the manuscript. Because of the comments from both the Editor and the Reviewers, we have made significant changes and have rewritten parts of the manuscript. Point to point responses to all comments are as follows.

Reviewer: 2

The paper " Intuitionistic Fuzzy Entropy for Group Decision Making of Water Engineering Project Delivery System Selection" addresses the problem of Group Decision Making through a methodology based on the Intuitionistic Fuzzy Entropy approach for the assessment of the weight of influencing factor of transaction modes 
and the weight of decision experts in a Delivery System Selection. Nevertheless, it has the following problems: Forms (Typo): The paper can´t published in its current state due to different kind of mistakes ranging from 

i) Typo (v.gr lines 17,18,117, 156, 170..),

Response:

We appreciate reviewer’s kind reminder, we have carefully gone through the whole manuscript and revised all typo.

Line 17: “were” has been revised to “was”

Line 18: “were” has been revised to “was”

Line 109: we have revised format through the whole manuscript to ensure its consistent

Line 151: sentence has been revised into:“The above equation was described as the relative entropy of X relative to Y”

Line 170: line 170 has been revised into line 164: “Definition 7 [68]: The defined  was entropy of IFS:, therefore, the following can be calculated :”

ii) Loosen acronyms (v.gr lines 61,62)

Response:

We have carefully checked all acronyms appeared in the manuscript, and make sure that its full name is provided at the first time appearance. Please see line 58-66.

iii)complete loosen paragraphs or leading to misunderstandings (v.gr. lines 88-100,109-112,128-129,152-154,194-197,210-215).

Response:

Line 88-100: We appreciate the reviewer for pointing these comments. We have delete these contents, which was put in wrong place by mistake.

Line 109-112:the repeated description has been deleted and its now become line 101-104: “however, it is obvious that although there are differences in the emphasis and quantity of the existing PDS index system, the main influencing factors of PDS selection can be summarized into three categories of owner characteristics, project characteristics and external environment.” Please refer to line 101-104.

Line 128-129(Please refer to line 120-121):sentence has been revised into“A greater entropy represents more uncertain information in the single evaluation result of the decision expert, and thus,a smaller weight should be given to such an expert.”

Line 152-154:we have rewritten the paragraph and now it become into the following (line 142-149):

Definition 3 [66]: As for the two intuitionistic fuzzy numbers:and,and are scoring functions of and respectively; and are exact function of  and respectively, so:

(1)if,then;

(2)if, there are three situations:

     a), then;

      b), then ;

    c), then.

Line 194-197:we have revised the sentence into line 189-194: “According to the entropy theory, if the entropy value for each criterion is smaller across alternatives, it should provide decision-makers with the useful information. Therefore, the criterion should be assigned a bigger weight; otherwise, such a criterion will be judged unimportant by most decision-makers. In other words, such a criterion should be evaluated as a very small weight. If the information about weight  of the criterion  is completely unknown, the entropy weights for determining the criteria weight can be calculated as follows”

Line 210-215:we have rewritten the content and now it become line 207-211:

“In the process of group decision-making, it is of great significance to objectively determine the weight of experts for more reliable decision-making. The idea of cross entropy of intuitionistic fuzzy sets is that if the cross entropy between the two experts is smaller, namely, the difference between their scores is smaller, then the individual evaluation results are relatively good, and a larger weight is given; while on contrary, a smaller weight is given”

Technical Contribution:

iv) It is quite confusing in lines 194-197 how weight entropies of each attribute can provide a sound decision-making assessment when an analysis of suitable thresholds has ‘not been formally achieved. In my opinion, an interval preference relation is enough for decision makers to provide preference relation.

Response:

To determine the entropy weights with respect to a set of criteria represented by IFSs, Ye (2010) introduced an entropy weight model, which can be utilized to find the optimal criteria weights, and proposed an assessment formula for a weighted correlation coefficient between a particular alternative and the hypothetical ideal alternative. An entropy weight model is established to determine criteria weights when knowledge of weights is definitely unknown, in which case the criteria values take the form of intuitionistic fuzzy numbers (IFNs). In this study, the entropy weights method was used to assign weights to each indicator measured for the considered alternatives.

References was incorporated in manuscript:

    [69] Ye, J. (2010). Fuzzy decision-making method based on the weighted

correlation coefficient under intuitionistic fuzzy environment. European Journal of

Operational Research, 205(1), 202-204.

v) It seems that the proposed method is a simple rewrite of previous work. There are not too many actual contributions and novelties.

Response:

    We thank the reviewer for this comment. Previous researchers have carried out extensive studies on the choice and decision of PDS. Overall, the present methods are helpful to solve the problem of PDS decision to a certain extent, but the choice of project delivery mode is often made intuitively according to the past experience and knowledge of decision makers, as well as the information and data of the water engineering project. Group decision-making systems such as peer-to-peer (P2P) systems can be easily modeled as Fuzzy logic. This opened a new area of research in decision making utilizing Fuzzy Sets (FSs) starting with Type-1 FSs, Type-2 FSs, and finally with Intuitionistic Fuzzy Sets. The comprehensive decision for project delivery is essentially a fuzzy multi-attribute group decision, many researchers have done a lot work on fuzzy decision making for PDS selection, but the characteristics and fuzziness of expert group were not considered yet.

The novelty of this research is basically to propose a generic 12-step fuzzy multi-criteria decision making (MCDM) method to select decision-making problems for water engineering project delivery system using a combination of three different techniques: (1) an intuitionistic fuzzy entropy method to identify the individual importance of criteria; (2) an IFHA operator to establish a sub-decision matrix with the weights; and (3) an IFWA operator to build a super-decision matrix with the weights. The present study aims to develop a more accurate and reliable PDS selection method. Meanwhile, we conducted a practical case analysis of a hydropower station to further demonstrated the feasibility, objectivity and scientific nature of the decision model, which has not been conducted before.

vi). The comparison section is incomplete. The authors are suggested to make comparisons with the state-of-the-art improving methods. It is difficult to evaluate the advantages of the proposed method. vii) One could expect the contribution to be based on the PDS, however, the results obtained are not contrasted against other methodologies.

Response:

In the present study, we are not actually doing simply comparisons but tried to introduce fuzzy entropy theory, a group decision making model to support PDS selection to reduce the judgment information losing and improve the objectivity and fairness of group decision-making. The intuitionistic fuzzy hybrid average (IFHA) operator and intuitionistic fuzzy weighted average (IFWA) operator on PDS decision are addressed to calculate the final decision weights. The present study aims to develop a more reliable PDS selection method. Meanwhile, we conducted a practical case analysis of a hydropower station to further demonstrated the feasibility, objectivity and scientific nature of the decision model, which has not been studied before.

Reviewer 3 Report

This article is built around the statement stated in line # 37. The authors must justify (either by references or by their own outlined argument). This needs to be clearly addressed by the authors.

There is a gap in the introduction section. There is no clear connection (in the paper) between the group decision-making systems and fuzzy logic. Therefore, I strongly urge the authors to add the following paragraph after line # 73.

-------------------------------------------------------------------------------------------------------

Group decision-making systems such as peer-to-peer (P2P) systems can be easily modeled as Fuzzy logic [reference #1, reference #2, reference #3]. This opened a new area of research in decision making utilizing Fuzzy Sets (FSs) starting with Type-1 FSs, Type-2 FSs, and finally with Intuitionistic Fuzzy Sets [reference #2].

References:

1.      Azzedin, F., Ridha, A., Rizvi, A.: Fuzzy trust for peer-to-peer based systems. In Proc. of World Academy of Science, Engineering and Technology, Vol. 21, 123127. (2007)

2.      Castillo O., Atanassov K. (2019) Comments on Fuzzy Sets, Interval Type-2 Fuzzy Sets, General Type-2 Fuzzy Sets and Intuitionistic Fuzzy Sets. In: Melliani S., Castillo O. (eds) Recent Advances in Intuitionistic Fuzzy Logic Systems. Studies in Fuzziness and Soft Computing, vol 372. Springer, Cham.

3.      Mostafavi, A. and M. Karamouz, Selecting appropriate project delivery system: Fuzzy approach with risk 379 analysis. Journal of Construction Engineering and Management, 2010. 136(8): 923-930. Atanassov, K.T., Intuitionistic fuzzy sets. Fuzzy Sets and Systems, 1986. 20(1): p. 87-96.

-------------------------------------------------------------------------------------------------------

English language and style:

Line # 88 starting from "2"  until line # 100: should be removed.

Line # 86: "dealt" should be changed to "deals".

Author Response

Response to the Editor and Reviewers’ Comments

Manuscript ID: entropy-628521

Title: "Intuitionistic Fuzzy Entropy for Group Decision Making of Water Engineering Project Delivery System Selection"

Author(s): XUN LIU *, FEI QIAN, LINGNA LIN, KUN ZHANG, LIANBO ZHU

Revision due before: 10-Nov-2019

We appreciate the time and effort of the Reviewers and the Editor in reviewing our manuscript. The reviews are very helpful for us to improve the manuscript. Because of the comments from both the Editor and the Reviewers, we have made significant changes and have rewritten parts of the manuscript. Point to point responses to all comments are as follows.

Reviewer: 3

General Comment:

This article is built around the statement stated in line # 37. The authors must justify (either by references or by their own outlined argument). This needs to be clearly addressed by the authors. There is a gap in the introduction section. There is no clear connection (in the paper) between the group decision-making systems and fuzzy logic. Therefore, I strongly urge the authors to add the following paragraph after line # 73.

-------------------------------------------------------------------------------------------------------

Group decision-making systems such as peer-to-peer (P2P) systems can be easily modeled as Fuzzy logic [reference #1, reference #2, reference #3]. This opened a new area of research in decision making utilizing Fuzzy Sets (FSs) starting with Type-1 FSs, Type-2 FSs, and finally with Intuitionistic Fuzzy Sets [reference #2].

References:

Azzedin, F., Ridha, A., Rizvi, A.: Fuzzy trust for peer-to-peer based systems. In Proc. of World Academy of Science, Engineering and Technology, Vol. 21, 123127. (2007) Castillo O., Atanassov K. (2019) Comments on Fuzzy Sets, Interval Type-2 Fuzzy Sets, General Type-2 Fuzzy Sets and Intuitionistic Fuzzy Sets. In: Melliani S., Castillo O. (eds) Recent Advances in Intuitionistic Fuzzy Logic Systems. Studies in Fuzziness and Soft Computing, vol 372. Springer, Cham. Mostafavi, A. and M. Karamouz, Selecting appropriate project delivery system: Fuzzy approach with risk 379 analysis. Journal of Construction Engineering and Management, 2010. 136(8): 923-930. Atanassov, K.T., Intuitionistic fuzzy sets. Fuzzy Sets and Systems, 1986. 20(1): p. 87-96.

Response:

We appreciate reviewer’s comment and suggestion.

We have added the following content in our manuscript, please see line 76-78: “Group decision-making systems such as peer-to-peer (P2P) systems can be easily modeled as Fuzzy logic [19-21]. This opened a new area of research in decision making utilizing Fuzzy Sets (FSs) starting with Type-1 FSs, Type-2 FSs, and finally with Intuitionistic Fuzzy Sets [20].”

Relevant references were added as well:

    [19] Azzedin, F., A. Ridha, and A. Rizvi. Fuzzy trust for peer-to-peer based

    systems. in Proceedings of World Academy of Science, Engineering and Technology. 2007.

    [20] Castillo, O. and K. Atanassov, Comments on fuzzy sets, interval type-2

    fuzzy sets, general type-2 fuzzy sets and intuitionistic fuzzy sets, in Recent Advances in Intuitionistic Fuzzy Logic Systems. 2019, Springer. 35-43.

    [21] Mostafavi, A. and M. Karamouz, Selecting appropriate project delivery system: Fuzzy approach with risk analysis. Journal of Construction Engineering and Management, 2010. 136(8): 923-930.

Reviewer 4 Report

The paper spends a lot of time describing the fuzzy logic and mathematical construct of the method but the application to water projects is rather cursory.  The value is in the mathematical derivation, and that relies on previous references.  So its really a paper about mathematical algorithm to inform decisions involving fuzzy logic.  One way to improve this paper would be to add at least one more and maybe two more applications, such as the same project delivery frameworks evaluated for a renewable energy project and and a civil insfrastructure project. 

Overall, I think this is a quality paper and useful contribution to the literature.

Author Response

Response to the Editor and Reviewers’ Comments

Manuscript ID: entropy-628521

Title: "Intuitionistic Fuzzy Entropy for Group Decision Making of Water Engineering Project Delivery System Selection"

Author(s): XUN LIU *, FEI QIAN, LINGNA LIN, KUN ZHANG, LIANBO ZHU

Revision due before: 10-Nov-2019

We appreciate the time and effort of the Reviewers and the Editor in reviewing our manuscript. The reviews are very helpful for us to improve the manuscript. Because of the comments from both the Editor and the Reviewers, we have made significant changes and have rewritten parts of the manuscript. Point to point responses to all comments are as follows.

Reviewer: 4

The paper spends a lot of time describing the fuzzy logic and mathematical construct of the method but the application to water projects is rather cursory.  The value is in the mathematical derivation, and that relies on previous references. So it’s really a paper about mathematical algorithm to inform decisions involving fuzzy logic.  One way to improve this paper would be to add at least one more and maybe two more applications, such as the same project delivery frameworks evaluated for a renewable energy project and a civil infrastructure project. Overall, I think this is a quality paper and useful contribution to the literature.

Response:

We thank the reviewer’s comment and suggestion. The main objective of the

present study was to establish a comprehensive scheme ranking model based on intuitionistic fuzzy hybrid average (IFHA) operator and intuitionistic fuzzy weighted average (IFWA) operator, and we demonstrated the model with a practical case analysis, we strongly agree that it would be meaningful to add one or more different applications of renewable energy project or civil infrastructure project, but considering words limit, we only put one case study in the present research, and the case analysis was enough to demonstrate and evaluate the addressed model.

Round 2

Reviewer 2 Report

I recommend to accept the manuscript after revisions have been made.